**Brief Communication**

# Organ Mapping Antibody Panels: a community resource for standardized multiplexed tissue imaging

Ellen M. Quardokus [1,38], Diane C. Saunders [2,3,38], Elizabeth McDonough [4], John W. Hickey [5], Christopher Werlein [6], Christine Surrette[4], Presha Rajbhandari[7], Anna Martinez Casals[5,8,9], Hua Tian [10], Lisa Lowery[4], Elizabeth K. Neumann[11,12,36], Frida Björklund[8], Taruna V. Neelakantan[13], Josh Croteau [14], Anne E. Wiblin [15], Jeremy Fisher [16], April J. Livengood[17], Karen G. Dowell[18], Jonathan C. Silverstein [19], Jeffrey M. Spraggins [11,12,20,21], Gloria S. Pryhuber [22], Gail Deutsch[23], Fiona Ginty[4], Garry P. Nolan[5], Simon Melov [24], Danny Jonigk [6,25,26], Michael A. Caldwell[27], Ioannis S. Vlachos [28,29,30], Werner Muller [31,32], Nils Gehlenborg [33], Brent R. Stockwell [7,13], Emma Lundberg [5,8,9], Michael P. Snyder [34], Ronald N. Germain [35], Jeannie M. Camarillo[27,37], Neil L. Kelleher [27,38], Katy Börner [1,38] & Andrea J. Radtke [35] ✉

Multiplexed antibody-based imaging enables the detailed characterization of molecular and cellular organization in tissues. Advances in the field now allow high-parameter data collection (>60 targets); however, considerable expertise and capital are needed to construct the antibody panels employed by these methods. Organ mapping antibody panels are community-validated resources that save time and money, increase reproducibility, accelerate discovery and support the construction of a Human Reference Atlas.

Multiplexed antibody-based imaging provides critical spatial data for mapping the vast network of cell types and anatomical structures present in multicellular organisms. Beyond preserving cell–cell interactions and tissue architecture, this approach offers insight into the cellular morphology and spatial patterns of complex tissues. When coupled with advanced analytical methods, high-content imaging allows for the quantification of heterogeneous cell types, including rare and difficult to extract populations. While imaging methods may vary in conjugate, mode of imaging or mode of immunolabeling, all aim for the in situ detection of molecular targets[1]. Importantly, these techniques are central to research efforts across several domains, but also foundational to international efforts aimed at building atlases of normal and diseased tissues.

Spatial mapping approaches pose substantial challenges as they are (1) targeted (antibodies must be carefully selected before data acquisition), (2) fallible (nonreproducible and off-target labeling are

well described[2,3]) (3) resource-intensive (a collection of 50 unique antibodies may require tens of thousands of US dollars in reagent costs and often months to build) and (4) dependent on subject matter experts for their construction and optimization[1].

To overcome these challenges, we are establishing a framework for the construction of organ mapping antibody panels (OMAPs)—combinations of antibodies that define cell populations and anatomical structures reproducibly in diverse tissues of human origin. This initiative emerged from the Human BioMolecular Atlas Program (HuBMAP)[4] and parallel efforts in the field of cytometry to construct peer reviewed optimized multicolor immunofluorescence panels (OMIPs)[5]. OMAPs expand upon other antibody validation efforts, such as the HuBMAP antibody validation reports and the Human Protein Atlas, by providing experimental details relevant for their successful application and domain expertise for atlas construction.

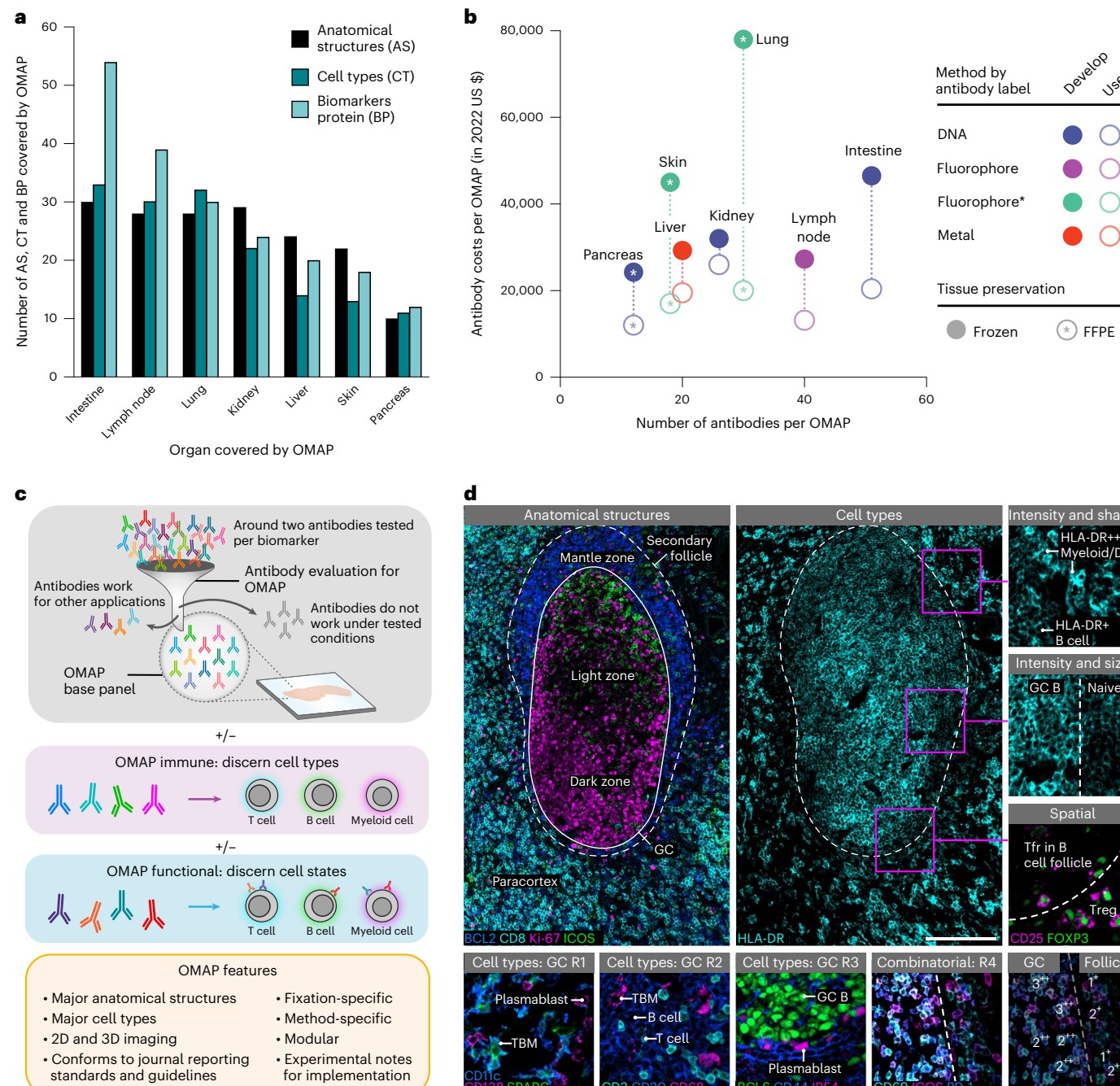

**Fig. 1 | OMAPs enhance standardization, discovery and stewardship of resources used in the spatial mapping of tissues at single-cell resolution.** **a**, Bar graph depicting the number of AS, CT and BP covered by each OMAP. Numbers were calculated by comparing with the relevant ASCT+B table[6] and were reviewed by domain experts. **b**, Plot showing the cost to develop (filled circles) and use (unfilled circle) the OMAPs described here. Costs (in 2022 US dollars) are shown for antibodies and/or conjugation kits and exclude labor, amortization of purchased equipment such as microscopes, and consumables other than antibodies and conjugation kits. Cost to use is calculated for 100 tests. DNA (CODEX), Fluorophore (IBEX), Fluorophore* (Cell DIVE), Metal (SIMS). **c**, Schematic detailing the creation of a base OMAP panel. Gray antibodies indicate antibodies that do not pass the first quality control check for accurate immunolabeling. Colored antibodies outside of the filter reflect validated antibodies that are not suitable for the final OMAP, for example, need amplification or a different conjugate. **d**, Images of human lymph node depicting AS, CT and cell states identified using IBEX. GC, Tfr (follicular regulatory T cells), Treg (regulatory T cells), TBM (tingible body macrophages) and R1–R4 refer to different regions in the secondary follicle. 3++ indicates CD69++, ICOS++ and PD-1++; 1+ indicates CD69+ or ICOS+ or PD-1+. Large insets, 150 μm; small insets, 50 μm. Representative of a dataset of ten samples.

OMAPs are tested rigorously to overcome technical challenges, such as steric hindrance, epitope loss, spectral overlap, target specificity and native tissue autofluorescence. Furthermore, OMAPs include details such as (1) critical markers for downstream analyses, (2) rationale for selected reagents, (3) four to six core markers to accommodate more traditional imaging techniques and (4) relevant details for implementation. OMAPs are designed for integration with the anatomical structures, cell types, plus biomarkers (ASCT+B) Reporter[6]—a state-of-the-art visualization tool (https://hubmapconsortium.github.io/ccf-asct-reporter/)—to facilitate tissue mapping efforts within and

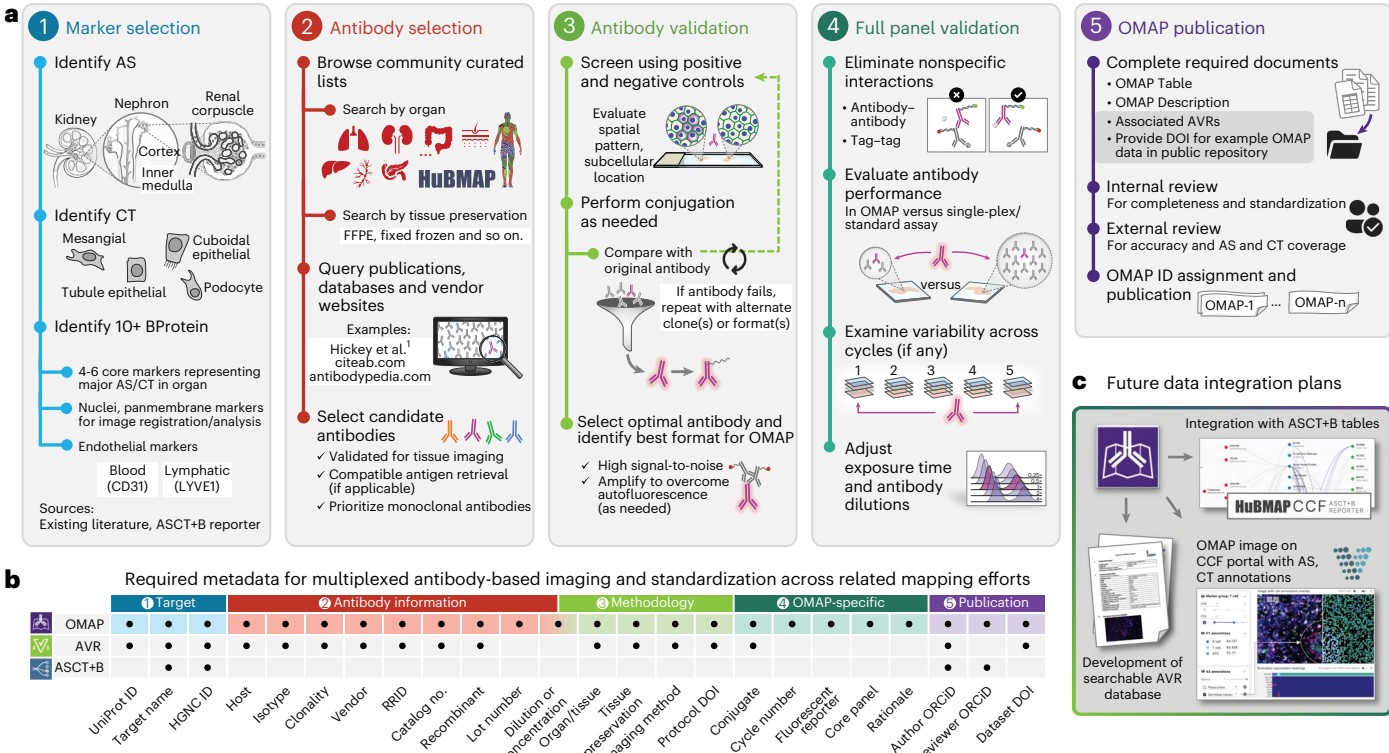

**Fig. 2 | Process, required metadata, and future data integration plans for OMAP effort. a**, Main milestones in constructing an OMAP, outlined via steps 1–5. Nephron and renal corpuscle illustrations were adapted from the HuBMAP CCF 2D Reference Object Library[21]. **b**, Metadata table illustrates the common fields shared across HuBMAP entities (OMAP, AVR and ASCT+B) to enhance search capabilities and promote community adoption of controlled vocabularies. Visit the Human Reference Atlas Portal (https://humanatlas.io/) for more details. **c**, In the future, OMAPs will be linked directly to the AVR searchable database, such that users can easily query information about BProtein markers for a specific OMAP. The ARWG is also collaborating with the Vitessce[19] software team to create a user interface with grouped image channels by BProtein and CT that additionally displays annotations by subject matter experts.

beyond the HuBMAP community. To this end, we strongly encourage inclusion of blood endothelial markers to empower construction of a Human Reference Atlas using the vasculature common coordinate framework (VCCF)[7] as well as lymphatic endothelial markers to further our understanding of the human lymphatic system[8]. We additionally recommend panels designed to evaluate signaling pathways, probe tissue-specific immunity and detect cell death processes under physiological and pathological conditions.

Here, we present an inaugural collection of OMAPs that provide a spatial context for 171 anatomical structures and 155 cell types in seven human organs, using 203 validated antibodies (Fig. 1a and Supplementary Tables 1 and 2). The described OMAPs represent multiple imaging modalities employing diverse antibody labels (DNA, fluorophore, metal), including codetection by indexing (CODEX)[9], iterative bleaching extends multiplexity (IBEX)[10], Cell DIVE[11] and secondary ion mass spectrometry (SIMS)[12]. Using data contributed by domain experts, we highlight several challenges related to the building of multiplexed panels while underscoring the value of developed OMAPs (Fig. 1b,c). First, an average of two antibodies were evaluated for each protein marker across all OMAPs, with some investigators screening multiple clones per target to ensure the best performing antibody was selected (around three per biomarker for Cell DIVE). The requirement to evaluate multiple clones and/or antibody formats is directly responsible for the substantial difference between the cost to design a new OMAP and the cost to use an existing one (Fig. 1b). Thus, we envision OMAPs serving as base panels that can be extended with curated marker sets in a modular fashion, saving researchers time and reagent costs (Fig. 1c). Lastly, even a sixplex panel can capture 63 theoretical cell types based on the presence or absence of a particular marker.

Such binary estimates undervalue the additional spatial, morphological and expression level differences that are critical for discerning structures, cell types and cell states in intact tissues (Fig. 1d and Supplementary Table 2).

OMAP construction begins with identifying the main anatomical structures (AS) and cell types (CT) present in a particular tissue or organ. Then, key protein biomarkers (BP) that characterize cell types of interest are determined (step 1; Fig. 2a). At a minimum, an OMAP should include at least ten protein targets along with critical markers for downstream image analysis (for example, nuclear and panmembrane markers for cell segmentation). Once a list of protein biomarkers is compiled, it is important to select antibodies compatible with the specific organ, tissue preservation method and multiplexed imaging platform (step 2, Fig. 2a). Presently, this process is achieved by querying antibody databases, existing literature and vendor websites for suitable candidates[1]. However, our community effort seeks to establish lists of expertly curated clones already validated for multiplexed tissue imaging, accelerating the selection process while establishing consensus among investigators (Supplementary Table 1). Another aim of this initiative is to support integration across tools and multimodal datasets generated by HuBMAP and other consortia for single-cell mapping. Accordingly, we report well-established gene and protein identifiers for each target using the HUGO Gene Nomenclature Committee (HGNC)[13] and Universal Protein Resource (UniProt) IDs[14] (Fig. 2b). OMAPs are linked to ASCT+B tables through their common metadata fields specifying each protein biomarker—making it easy to map experimental data to the evolving Human Reference Atlas.

Following antibody selection, each antibody is extensively characterized before inclusion in an OMAP (step 3; Fig. 2a). Several approaches

for validating antibodies have been described: positive and negative controls, colocalization with orthogonal markers and assessing the spatial pattern and subcellular localization of antibody labeling based on published data[1–3]. Importantly, these practices and relevant metadata are captured in antibody validation reports (AVRs)—a parallel effort that complements the OMAP initiative (Supplementary Table 3). AVRs provide an overview of the validated antibodies for each protein marker, any alternative clones tested, specific details on the characterization process and representative images to allow qualitative assessment of antibody specificity, sensitivity and reproducibility. AVRs and OMAPs conform to antibody reporting guidelines designed to thwart the replication crisis in biomedical research by including fields that uniquely identify a reagent, such as a research resource identifier (RRID)[15]. In addition to capturing antibody-specific information, AVRs and OMAPs also include links to relevant protocols and critical methodology details (Fig. 2b). Beginning in 2023, all OMAP authors will be asked to contribute an AVR for each antibody included in their panel. Future goals include data integration between AVRs and OMAPs to support OMAP construction using well-characterized reagents.

The next step of OMAP construction is validating the full panel by assessing nonspecific interactions, spectral overlap and potential impact of cycle number on immunogenicity and tissue loss[1,10,16,17] (step 4; Fig. 2a). Several antibodies, reflecting different clones and/or conjugates, may be evaluated and compared with their performance in traditional imaging assays and serial sections. Each antibody must be carefully titrated and exposure times adjusted to yield the best signal-to-noise for a given antibody. These details are included in the supporting materials required for each OMAP: the OMAP Table (Supplementary Table 4), OMAP Description Document (Supplementary Table 5) and AVRs in the next release (step 5; Fig. 2a). In contrast to AVRs, OMAPs also report the cell markers that characterize distinct cell types and states in different human tissues. These designations are assigned by OMAP authors, reviewed by subject matter experts, and annotated using standardized cell ontology (CL) terms[6,18], allowing for future integration with the ASCT+B Reporter (Supplementary Table 2). The rationale for including a particular antibody is documented in the OMAP Table provided by the contributing author(s) (Supplementary Table 4). After review by subject matter experts, OMAPs are given a digital object identifier (DOI) and published online with a representative dataset deposited into a public repository—a requirement starting in 2023. A high priority within the coming year is expanding the functionality of Vitessce—an open-source interactive visualization framework for exploration of multimodal and spatially resolved single-cell data[19]— to allow visualization of imaging datasets with expert annotations for anatomical structures and cell types (Fig. 2c).

By establishing OMAPs, we aim to offset the considerable time and cost associated with creating such resources de novo, while standardizing data acquisition and reporting for multiplexed tissue imaging studies—a key objective in the field[20]. To achieve this goal, we invite the spatial biology community to construct OMAPs for use in 2D and 3D imaging of healthy, diseased and aging tissues, such as those acquired through the SenNet program (https://sennetconsortium.org). Beyond conforming to journal reporting guidelines, OMAPs and associated AVRs establish confidence in antibody clones by aggregating usage data across laboratories and technologies. Data from studies that use OMAPs are automatically aligned to, and can be compared with, data in the Human Reference Atlas[6], providing evidence for cell types in specific anatomical structures. In closing, OMAPs save time and money, increase reproducibility, support Human Reference Atlas construction, and accelerate biological insights gained from multiplexed tissue imaging.

## Online content

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

[1]Department of Intelligent Systems Engineering, Indiana University, Bloomington, IN, USA. [2]Department of Medicine, Vanderbilt University Medical Center, Nashville, TN, USA. [3]Vanderbilt Diabetes Center, Vanderbilt University School of Medicine, Nashville, TN, USA. [4]GE Research, Niskayuna, NY, USA. [5]Department of Pathology, Stanford University School of Medicine, Stanford, CA, USA. [6]Institute of Pathology, Hannover Medical School, Hannover, Germany. [7]Department of Biological Sciences, Columbia University, New York, NY, USA. [8]Science for Life Laboratory, School of Engineering Sciences in Chemistry, Biotechnology and Health, KTH Royal Institute of Technology, Stockholm, Sweden. [9]Department of Bioengineering, Stanford University, Stanford, CA, USA. [10]Department of Chemistry, Pennsylvania State University, University Park, PA, USA. [11]Department of Biochemistry, Vanderbilt University, Nashville, TN, USA. [12]Mass Spectrometry Research Center, Vanderbilt University, Nashville, TN, USA. [13]Department of Chemistry, Columbia University, New York, NY, USA. [14]Department of Business Development, BioLegend Inc., San Diego, CA, USA. [15]Department of Research and Development, Abcam PLC, Discovery Drive, Cambridge Biomedical Campus, Cambridge, UK. [16]Department of Research and Development, Cell Signaling Technology, Inc., Danvers, MA, USA. [17]Department of Protein and Cell Analysis, Thermo Fisher Scientific, Carlsbad, CA, USA. [18]Akoya Biosciences, Marlborough, MA, USA. [19]Department of Biomedical Informatics, University of Pittsburgh School of Medicine, Pittsburgh, PA, USA. [20]Department of Chemistry, Vanderbilt University, Nashville, TN, USA. [21]Department of Cell and Developmental Biology, Vanderbilt University, Nashville, TN, USA. [22]Department of Pediatrics, University of Rochester Medical Center, Rochester, NY, USA. [23]Department of Pathology, University of Washington Medical Center, Seattle, WA, USA. [24]Buck Institute for Research on Aging, Novato, CA, USA. [25]German Center for Lung Research (DZL), Biomedical Research in Endstage and Obstructive Lung Disease Hannover (BREATH), Hannover, Germany. [26]Institute of Pathology, RWTH University of Aachen, Aachen, Germany. [27]Departments of Chemistry, Molecular Biosciences and the Proteomics Center of Excellence, Northwestern University, Evanston, IL, USA. [28]Spatial Technologies Unit, Harvard Medical School Initiative for RNA Medicine, Beth Israel Deaconess Medical Center, Boston, MA, USA. [29]Department of Pathology, Beth Israel Deaconess Medical Center, Boston, MA, USA. [30]Broad Institute of MIT and Harvard, Cambridge, MA, USA. [31]Miltenyi Biotec B.V. and Co. KG, Bergisch Gladbach, Germany. [32]Division of Infection, Immunity and Respiratory Medicine, University of Manchester, Manchester, UK. [33]Department of Biomedical Informatics, Harvard Medical School, Boston, MA, USA. [34]Department of Genetics, Stanford University School of Medicine, Stanford, CA, USA. [35]Laboratory of Immune System Biology, Lymphocyte Biology Section and Center for Advanced Tissue Imaging, NIAID, NIH, Bethesda, MD, USA. [36]Present address: Department of Chemistry, University of California Davis, Davis, CA, USA. [37]Present address: Inotiv, Nashville, TN, USA. [38]These authors contributed equally: Ellen M. Quardokus, Diane C. Saunders, Neil Kelleher and Katy Börner. ✉e-mail: andrea.radtke@nih.gov

## Methods

### Overview

All OMAPs require completion of an OMAP Table and an OMAP Description Document. Additional details on how to complete these documents are included in Supplementary Tables 4 and 5. Beginning in 2023, each OMAP will require a representative dataset deposited to a public repository and an AVR for each included antibody. Additional details related to the construction of OMAPs can be found in our standard operating procedure (SOP)[22] and frequently asked questions (FAQs) on the Human Reference Atlas Portal: https://humanatlas.io/omap.

### Marker selection

The first step in OMAP construction is to identify the main anatomical structures and cell types for an organ of interest. This will inform the biomarkers to target using appropriate antibodies. The ASCT+B Reporter[6] (https://hubmapconsortium.github.io/ccf-asct-reporter/) is a useful resource reporting the AS, CT and gene and protein BP for several human organs. To obtain a spreadsheet of the AS and CT present in an organ of interest, select the newest version of an organ-specific ASCT+B table and use the 'Report' feature to download a spreadsheet listing the AS and CT. The best way to view the BP for a particular cell type is to visit the Data Tables in the ASCT+B Reporter. The biomarkers used to phenotype a particular cell are listed sequentially as BProtein/1, BProtein/2 and BProtein/3. Additional resources for identifying cell markers are described in the OMAP SOP[22] and a multiplexed tissue imaging primer[1]. Authors should designate four to six markers that, when used together, allow profiling of main anatomical structures and cell types in a given tissue. Beyond these core markers, an OMAP should allow ten or more unique biomarkers to be visualized in a single tissue section and support downstream image analysis with appropriate nuclear and panmembrane targets. The inclusion of antibodies directed against one or more blood endothelial markers (for example, CD31) is strongly encouraged to support the construction of a Human Reference Atlas using the Vasculature Common Coordinate Framework (VCCF)[7,23,24]. Additionally, antibodies directed against one or more lymphatic endothelial markers (for example, LYVE1) are highly recommended to further our understanding of the human lymphatic system[8].

### Antibody selection and validation

Resources for antibody selection include existing OMAPs (https://humanatlas.io/omap), antibody search engines, an extensive clone list included in a multiplexed imaging primer[1] and Supplementary Tables 1 and 2. Using these resources, antibodies can be selected for a desired tissue preservation method, imaging platform and universal antigen retrieval conditions, if applicable. In general, antibodies validated by vendors for immunohistochemistry (IHC) of formalin-fixed paraffin-embedded (FFPE) samples often work as fluorophore-conjugated antibodies for FFPE or tissues fixed using 1–4% paraformaldehyde. Each antibody included in a published OMAP must be validated using well-described practices[1–3], for example, evaluating the immunolabeling pattern of a particular antibody using positive and negative controls and colocalization with orthogonal markers.

### Full panel (OMAP) validation

Following selection of individual antibodies, the full panel of antibodies must be validated as an assembly. Several excellent resources are available on the process of panel construction and additionally include validated panels for diverse human tissues[1,16,25,26]. It is difficult to generalize across platforms, as differences exist between methods employing fluorophore- versus metal-labeled antibodies and techniques using cyclic or all-in-one imaging[1]. Nevertheless, several OMAP validation steps are shared across multiplexed imaging methods. First, the performance of an antibody in an OMAP must be compared with its performance in a single-plex assay using qualitative and quantitative assessments, for example, spatial pattern, subcellular location and signal intensity. Second, nonspecific interactions, such as cross-reactivity between antibodies and tag–tag interactions, should be evaluated and eliminated by selecting distinct clones, conjugating to other oligonucleotide tags, and moving antibodies to a different cycle depending on the overall panel design. Last, each antibody in an OMAP must be carefully selected to yield a high signal-to-noise ratio and titrated to eliminate nonspecific binding and minimize spectral overlap, if applicable. Cyclic methods employing fluorescent antibodies or reporters additionally need to evaluate the impact of cycle number on antibody staining quality and tissue integrity. Detailed protocols and examples on how to evaluate the impact of cycle number on immunogenicity and tissue loss have been reported[10,16,17].

### OMAP review and publication

A chief aim of this work is to create multiplexed imaging panels that can be used across laboratories to generate high-quality spatial data from human tissues. To achieve this aim, it is imperative that the described OMAPs are reviewed by experts in pathology, histology, cell biology and/or multiplexed imaging. OMAP authors must perform an internal review to ensure that all required documents are completed according to established guidelines[22] (Supplementary Tables 4 and 5). Next, OMAP authors need to quantify the number of AS and CT profiled by an OMAP using the relevant ASCT+B table for their organ and other literature. Furthermore, the biomarkers used to define each cell type must be documented as described in the SOP and shown in Supplementary Table 2. Before publication, OMAPs are additionally reviewed by pathologists and members of the HuBMAP consortium. The external review process includes evaluating a prospective OMAP for completeness, accuracy, standardization with existing OMAPs and coverage of anatomical structures, cell types and biomarkers. Once the review process is complete, OMAPs are assigned a unique number that reflects the date created and version of the OMAP. OMAPs are also given a DOI for citation purposes and published online on the Human Reference Atlas Portal: https://humanatlas.io/omap.

### Governance

HuBMAP and 16 international consortia are collaborating on the construction of a Human Reference Atlas[6]. Experts across these consortia are organized via the Anatomical Structures, Cell Types and Biomarkers Working Group (ASCT+B WG) that meets monthly to discuss datasets, software and cross-consortia efforts such as OMAPs. OMAPs are critically important for the construction of a Human Reference Atlas as they contain knowledge on cell types and protein biomarkers for spatial mapping of human organs. The Human Reference Atlas has a 6-month release cycle that includes the publication of new, versioned datasets for AS, CT and biomarkers (ASCT+B) tables, OMAPs and three-dimensional (3D) reference organs. Additional meetings are scheduled as needed to resolve conflicts and advance new work. The Affinity Reagent Imaging and Validation Working Group (ARWG) meets monthly to discuss topics related to the field of multiplexed imaging. The ARWG is focused on the construction, review and publication of AVRs and OMAPs. Members of both the ASCT+B WG and ARWG advise on suggested updates and adjudicate disagreements among OMAP authors and reviewers. These conversations will be facilitated by the personnel listed in the OMAP SOP[22]. The longevity and continuity of the OMAP effort will be achieved through engagement with dozens of consortia and cross-training members to perform different roles required for OMAP review, publication and usage.

### Reporting summary

Further information on research design is available in the Nature Portfolio Reporting Summary linked to this article.

## Data availability

The datasets described in this manuscript are publicly available on the Human Reference Atlas Portal: https://humanatlas.io/omap.

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

## Acknowledgements

We are grateful for engaging and thoughtful discussions from the Affinity Reagent Imaging and Validation Working Group and HuBMAP Consortium. We wish to thank A. Verdoes and B. Herr II for project management and computational support, respectively. The authors would like to acknowledge funding from the following sources: DK20593 (D.C.S.); National Institutes of Health (NIH) U54 DK134302, NIH U54 EY032442, NIH R01 AG078803, NIH U01 DK133766 (J.M.S.); NIH T32ES007028 and F32DK128887 (E.K.N.); NIH U54 HG010426-01 (M.P.S. and G.P.N.); NIH UG3 HL145600-01, NIH UH3 CA246635-01 and P41 GM108569-07 (N.L.K., J.M.C. and M.A.C.); National Science Foundation (NSF) GRFP DGE-2036197 (T.V.N.); NIH 1UG3CA256962 and 4UH3CA256962 (B.R.S. and H.T.); NIH UH3 CA246594-01 (E.M., L.L., C.S. and F.G.); R35CA209896 (B.R.S.); EU Horizon 2020, 874710 and NIH 5U01DK120447 (E.L.); NIH T32CA196585 and ACS PF-20-032-01-CSM (J.W.H.); U54HL145608 (G.S.P. and G.D.); NIH U54 AG075932 and UG3 CA268105 (S.M.); U54 U54HL1654401 (I.S.V.); and U01HL148861-02S1 (G.S.P. and G.D.). C.W. and D.J. are supported by the grant of the European Research Council (ERC); European Consolidator Grant, XHale (Reference no. 771883). This work was supported, in part, by the Intramural Research Program of the NIH, National Institute of Allergy and Infectious Diseases (NIAID) and National Cancer Institute (NCI). This research has been funded by the NIH under Human Biomolecular Atlas Program (HuBMAP) awards OT2OD026671 (K.B. and E.M.Q.), OT2OD026675 (J.C.S), OT2OD033758 (N.G.), the Cellular Senescence Network (SenNet) Consortium Organization and Data Coordinating Center (CODCC) award U24CA268108 (K.B., E.M.Q., J.C.S. and N.G.). We thank J. Hernandez and J. Davis (NCI, NIH) for providing deidentified human lymph nodes featured in this work. We thank A. Bosio (Miltenyi Biotec B.V. and Co. KG) for the support of W.M.

## Author contributions

E.M.Q. contributed to the OMAP and AVR file formats, reviewed OMAPs, helped write the SOP and portal descriptions with A.J.R., K.B. and D.C.S. and published OMAPs on the HuBMAP CCF Portal. D.C.S. contributed to the OMAP and AVR file formats, reviewed OMAPs, contributed to the writing of the manuscript and designed figures and tables with A.J.R., J.W.H., A.M.C., P.R., C.W. and C.S. with support from J.M.S. J.W.H. authored the intestine OMAP and contributed to figure design and writing with support from G.P.N. E.M. authored the skin OMAP, created the AVR file format with N.K., J.M.C. and M.A.C., and contributed domain knowledge to the conception of the figure and manuscript. C.W. served as an expert reviewer for all OMAPs with W.M. and support from D.J. and created data tables with C.S. and A.J.R. P.R., T.V.N. and H.T. authored the liver OMAP, contributed data for the figures and tables and reviewed OMAPs with support from B.R.S. C.S. and L.L. authored the lung OMAP, created data tables and contributed data for the display items with support from G.S.P., G.D. and F.G. A.M.C. and F.B. authored the pancreas OMAP, contributed data for display items, and reviewed OMAPs with support from E.L. E.K.N. authored the kidney OMAP and contributed data for the figure with support from J.M.S. J.C., A.E.W., J.F., A.J.L., K.G.D. and W.M. offered technical insights and commercial domain knowledge. I.S.V. and D.J. provided technical insights and domain knowledge. J.C.S. designed and supported fields for integration of OMAPs with HuBMAP infrastructure. N.G. contributed OMAP data visualization strategies and software. S.M., M.P.S., R.N.G., N.L.K. and J.M.C contributed ideas and suggestions for the conception and design of AVRs and OMAPs. K.B. provided expertise on the integration of OMAPs with visualization tools, provided important edits to the OMAP and CCF portal descriptions, SOP and manuscript. A.J.R. contributed to the OMAP file format, authored the LN OMAP, contributed data for the figure, reviewed OMAPs, wrote the first draft of the manuscript, created the figures with D.C.S. and J.W.H. and authored the SOP and portal page descriptions with E.Q., K.B. and D.C.S. All authors contributed to the review and editing of this work. E.M., J.W.H., C.W., C.S., P.R., and A.M.C. contributed equally as co-second authors.

## Competing interests

A.E.W. is an employee and shareholder of Abcam PLC. J.F. is an employee of Cell Signaling Technology, Inc. J.C. is an employee and stakeholder of BioLegend. A.J.L. is an employee and shareholder of Thermo Fisher Scientific. E.M., L.L., C.S. and F.G. are employees of GE Research. B.R.S. is an inventor on patents and patent applications involving small molecule drug discovery, and the 3F3-FMA antibody, cofounded and serves as a consultant to Exarta Therapeutics and ProJenX Inc.; holds equity in Sonata Therapeutics, receives sponsored research from Sumitomo Pharma Oncology Inc and serves as a consultant to Weatherwax Biotechnologies Corporation and Akin Gump Strauss Hauer and Feld LLP. M.P.S. is cofounder and advisory board member of Personalis, Qbio, January AI, Mirvie, Filtricine, Fodsel, Lollo and Protos. G.P.N. has equity in and is a scientific advisory board member of Akoya Biosciences, Inc. I.S.V. consults for Guidepoint Global, Cowen, Mosaic and NextRNA. N.G. is a cofounder and equity owner of Datavisyn. W.M. is an employee of Miltenyi Biotec B.V. and Co. KG. E.L. serves as a consultant for 10x Genomics, Moleculent AB, Pixelgen Technologies and Nautilus Biotechnology. K.G.D. is an employee of Akoya Biosciences. The remaining authors declare no competing interests.

## Additional information

**Correspondence and requests for materials** should be addressed to Andrea J. Radtke.

# Reporting Summary

## Statistics

For all statistical analyses, confirm that the following items are present in the figure legend, table legend, main text, or Methods section.

| n/a | Confirmed | |
|---|---|---|
| ☐ | ☒ | The exact sample size (*n*) for each experimental group/condition, given as a discrete number and unit of measurement |
| ☒ | ☐ | A statement on whether measurements were taken from distinct samples or whether the same sample was measured repeatedly |
| ☒ | ☐ | The statistical test(s) used AND whether they are one- or two-sided<br>*Only common tests should be described solely by name; describe more complex techniques in the Methods section.* |
| ☒ | ☐ | A description of all covariates tested |
| ☒ | ☐ | A description of any assumptions or corrections, such as tests of normality and adjustment for multiple comparisons |
| ☒ | ☐ | A full description of the statistical parameters including central tendency (e.g. means) or other basic estimates (e.g. regression coefficient) AND variation (e.g. standard deviation) or associated estimates of uncertainty (e.g. confidence intervals) |
| ☒ | ☐ | For null hypothesis testing, the test statistic (e.g. *F*, *t*, *r*) with confidence intervals, effect sizes, degrees of freedom and *P* value noted<br>*Give P values as exact values whenever suitable.* |
| ☒ | ☐ | For Bayesian analysis, information on the choice of priors and Markov chain Monte Carlo settings |
| ☒ | ☐ | For hierarchical and complex designs, identification of the appropriate level for tests and full reporting of outcomes |
| ☒ | ☐ | Estimates of effect sizes (e.g. Cohen's *d*, Pearson's *r*), indicating how they were calculated |

*Our web collection on statistics for biologists contains articles on many of the points above.*

## Software and code

Policy information about availability of computer code

| Data collection | ASCT+B Reporter (v2.5.0; https://github.com/hubmapconsortium/ccf-asct-reporter/), supporting data from HRA CCF release v1.2 |
|---|---|
| Data analysis | *Provide a description of all commercial, open source and custom code used to analyse the data in this study, specifying the version used OR state that no software was used.* |

For manuscripts utilizing custom algorithms or software that are central to the research but not yet described in published literature, software must be made available to editors and reviewers. We strongly encourage code deposition in a community repository (e.g. GitHub). See the Nature Portfolio guidelines for submitting code & software for further information.

## Data

Policy information about availability of data

All manuscripts must include a data availability statement. This statement should provide the following information, where applicable:

- Accession codes, unique identifiers, or web links for publicly available datasets
- A description of any restrictions on data availability
- For clinical datasets or third party data, please ensure that the statement adheres to our policy

Data Availability: The datasets described in this manuscript are publicly available on the Human Reference Atlas Portal: https://humanatlas.io/omap.

# Human research participants

Policy information about studies involving human research participants and Sex and Gender in Research.

| | |
|---|---|
| Reporting on sex and gender | Not applicable |
| Population characteristics | Not applicable |
| Recruitment | Not applicable |
| Ethics oversight | Not applicable |

Note that full information on the approval of the study protocol must also be provided in the manuscript.

# Field-specific reporting

Please select the one below that is the best fit for your research. If you are not sure, read the appropriate sections before making your selection.

☒ Life sciences ☐ Behavioural & social sciences ☐ Ecological, evolutionary & environmental sciences

For a reference copy of the document with all sections, see nature.com/documents/nr-reporting-summary-flat.pdf

# Life sciences study design

All studies must disclose on these points even when the disclosure is negative.

| | |
|---|---|
| Sample size | This manuscript describes a community-validated resource for the field of spatial biology. It does not include experiments related to answering a biological question or developing a method. Therefore, sample size is not applicable. |
| Data exclusions | No data were excluded from this study. |
| Replication | Figure 1d presents imaging data generated by a collection of validated antibodies (OMAP). It was repeated more than 10 times and all attempts at replication were successful. |
| Randomization | Randomization is not applicable because this study does not report experimental data but instead describes a community initiative. |
| Blinding | Blinding is not applicable because this study does not report experimental data but instead describes a community initiative. |

# Reporting for specific materials, systems and methods

We require information from authors about some types of materials, experimental systems and methods used in many studies. Here, indicate whether each material, system or method listed is relevant to your study. If you are not sure if a list item applies to your research, read the appropriate section before selecting a response.

### Materials & experimental systems

| n/a | Involved in the study |
|---|---|
| ☐ | ☒ Antibodies |
| ☒ | ☐ Eukaryotic cell lines |
| ☒ | ☐ Palaeontology and archaeology |
| ☒ | ☐ Animals and other organisms |
| ☒ | ☐ Clinical data |
| ☒ | ☐ Dual use research of concern |

### Methods

| n/a | Involved in the study |
|---|---|
| ☒ | ☐ ChIP-seq |
| ☒ | ☐ Flow cytometry |
| ☒ | ☐ MRI-based neuroimaging |

## Antibodies

| | |
|---|---|
| Antibodies used | All relevant antibody details (catalog number, RRID, lot number, etc) are included in Supplementary Table 1 and shared publicly on the Human Reference Atlas Portal: https://humanatlas.io/omap. |
| Validation | Validation details for each antibody will be included in Antibody Validation Reports (AVRs) described in this work and posted online in 2023 on https://avr.hubmapconsortium.org/ (update in progress). |

