## [Peer Review File · Nature Methods]

Peer Review Information

Manuscript Title: Organ Mapping Antibody Panels (OMAPs): A community resource for standardized multiplexed tissue imaging

Corresponding author name(s): Andrea Radtke

Editorial Notes: n/a

Reviewer Comments & Decisions:

Decision Letter, initial version:

Dear Andrea,

Thank you for your letter detailing how you would respond to the reviewer concerns regarding your Correspondence, "Organ Mapping Antibody Panels (OMAPs): A community resource for standardized multiplexed tissue imaging".

We have decided to invite you to revise your manuscript as you have outlined, before we reach a final decision on publication. In particular, we suggest that you expand the Correspondence into a Brief Communication. This will allow you to add more detail on the OMAPs as detailed in your response to the reviewers. Brief Communications are research papers so please make sure to add some characterization data as well as a Methods and Supplementary Information section. Here is more information on the format type - <https://www.nature.com/nmeth/content>.

* include a point-by-point response to the reviewers and to any editorial suggestions

* please underline/highlight any additions to the text or areas with other significant changes to facilitate review of the revised manuscript

- * address the points listed described below to conform to our open science requirements
- * ensure it complies with our general format requirements as set out in our guide to authors at www.nature.com/naturemethods
- * resubmit all the necessary files electronically by using the link below to access your home page

[Redacted] This URL links to your confidential home page and associated information about manuscripts you may have submitted, or that you are reviewing for us. If you wish to forward this email to co-authors, please delete the link to your homepage.

We hope to receive your revised paper within 4 weeks. If you cannot send it within this time, please let us know. In this event, we will still be happy to reconsider your paper at a later date so long as nothing similar has been accepted for publication at Nature Methods or published elsewhere.

Sincerely,
Madhura

Madhura Mukhopadhyay, PhD
Senior Editor
Nature Methods

OPEN SCIENCE REQUIREMENTS

REPORTING SUMMARY AND EDITORIAL POLICY CHECKLISTS

Please note that these forms are dynamic ‘smart pdfs’ and must therefore be downloaded and completed in Adobe Reader. We will then flatten them for ease of use by the reviewers. If you would like to reference the guidance text as you complete the template, please access these flattened versions at <http://www.nature.com/authors/policies/availability.html>.

IMAGE INTEGRITY

DATA AVAILABILITY

Please include a “Data availability” subsection in the Online Methods. This section should inform readers about the availability of the data used to support the conclusions of your study, including accession codes to public repositories, references to source data that may be published alongside the paper, unique identifiers such as URLs to data repository entries, or data set DOIs, and any other statement about data availability. At a minimum, you should include the following statement: “The data that support the findings of this study are available from the corresponding author upon request”, describing which data is available upon request and mentioning any restrictions on availability. If DOIs are provided, please include these in the Reference list (authors, title, publisher (repository name),

identifier, year). For more guidance on how to write this section please see:

<http://www.nature.com/authors/policies/data/data-availability-statements-data-citations.pdf>

CODE AVAILABILITY

Please include a “Code Availability” subsection in the Online Methods which details how your custom code is made available. Only in rare cases (where code is not central to the main conclusions of the paper) is the statement “available upon request” allowed (and reasons should be specified).

MATERIALS AVAILABILITY

SUPPLEMENTARY PROTOCOL

To help facilitate reproducibility and uptake of your method, we ask you to prepare a step-by-step Supplementary Protocol for the method described in this paper. We [encourage authors to share their step-by-step experimental protocols](https://www.nature.com/nature-research/editorial-policies/reporting-standards#protocols) on a protocol sharing platform of their choice and report the protocol DOI in the reference list. Nature Portfolio's Protocol Exchange is a free-to-use and open resource for protocols; protocols deposited in Protocol Exchange are citable and can be linked from the published article. More details can found at www.nature.com/protocolexchange/about.

ORCID

Nature Methods is committed to improving transparency in authorship. As part of our efforts in this direction, we are now requesting that all authors identified as ‘corresponding author’ on published papers create and link their Open Researcher and Contributor Identifier (ORCID) with their account on the Manuscript Tracking System (MTS), prior to acceptance. This applies to primary research papers only. ORCID helps the scientific community achieve unambiguous attribution of all scholarly contributions. You can create and link your ORCID from the home page of the MTS by clicking on ‘Modify my Springer Nature account’. For more information please visit www.springernature.com/orcid.

--

Reviewers' Comments:

Reviewer #1:

Remarks to the Author:

This work is not a traditional research report, but rather outline of an idea that is underway to product Organ Mapping Antibody Panels (OMAPs). The concept addresses a key problem in the spatial biology field; the challenge of antibody-based cell identification. OMAP proposes generation and sharing of panels of antibodies to facilitate tissue mapping that can be universally used to define cell types and anatomic structures. This work is the announcement of the first data installment including 204 antibodies that identify 95 structures and 178 cell types.

Overall, this is a great concept. Atlases are useful but can show conflicting data. This proposed approach has the potential to minimize conflict by sharing data, images, methods, and reagents. While time is needed to prove its true value, this introduction seems promising and provocative.

Reviewer #2:

Remarks to the Author:

In the correspondence manuscript entitled “Organ Mapping Antibody Panels (OMAPs): A community resource for standardized multiplexed tissue imaging” (NMEMH-C48483A) an effort is presented for sharing information and standardizing antibody panels for use in multiplexed tissue imaging experiments. The correspondence identifies key challenges in the design and execution of these imaging studies. It appears that there are two key parts to the generation of each particular OMAP: 1. expert selection of targets that are deemed necessary for a deep characterization of a specific tissue type and 2. rigorous antibody testing to identify the optimal reagents to use for each particular technology. While a hub for antibody information and guidance on tissue-specific panels is needed to help overcome some

of the barriers for imaging experiments, the current instantiation of the OMAP project raises several concerns.

1. More clarity is needed about the criteria and quantitative measures that will be used for optimal reagent selection and rigorous reagent standardization. Review of the manuscript and the OMAP SOP available online does not provide a clear sense of the specific criteria that were used for selecting the antibodies that are included in the current OMAPs (versus the ones that were triaged as having inferior performance). Were similar criteria used by all groups or are the resulting first generation OMAPs based upon very different standards implemented by each of the groups? What criteria should others in the spatial biology community use to suggest changes, additions, improvements, etc. Presumably the best clone was evaluated by visual review of images and also by rigorous quantitative methods that permit the assessment of signal to noise and other features of antibody performance; such data is important for others to review/assess and a detailed process for antibody comparison is needed if the goal is rigorous antibody standardization. It is important to record the types of control tissues and samples that were used in these evaluations. Also, antibody clones may seem to perform differently when assessed with different analytical pipelines (thus, the analysis pipeline used for the assessments should be indicated). Moreover, it seems that only the 'best' antibodies that were selected for each OMAP are presented – it is equally important to know which antibodies were evaluated and deemed to be of lesser quality/value for the particular technology/application. Without reporting the specific quantitative criteria and process for selecting optimal antibodies, it is unclear how choices will be made as this resource develops and expands.

2. It seems that structured data will be necessary to maintain uniform reporting and information communication in the OMAP data tables. Even with the first seven OMAPs that have been generated by groups that work very closely together in the same consortium, there is a sense that complexity will quickly arise. Even among a relatively small number of groups, the presentation of the data needs improvement: Smooth muscle actin is called SMA, α -SMA, \pm Smooth Muscle Actin. Antibody use is recommended as either a dilution or a specific concentration ($\mu\text{g/ml}$). In the Skin OMAP, the recommended use is "Dilution/Concentration" and then a number is provided that is presumably the concentration. In the 'conjugate' column for the Intestine OMAP, 'oligonucleotide' is listed for all antibodies whereas for other OMAPs that also use CODEX a specific fluor is provided (which is the proper way of presenting the information). The rationale for the antibody used is not uniformly presented either (for example CD4 marks both T helper cells as well as macrophages at lower levels presumably in all tissues but that is the listed rationale for only some of the CD4 antibodies but not others). There are even a couple of antibodies that come from an individual investigator's lab which will not presumably be widely available to others. At the inception it will be important to have more structured ways of reporting and ingesting data otherwise these tables will quickly become difficult to use. Is structured information ingress planned?

3. While standardizing data ingress of structured data is ultimately sufficiently straightforward as a concept (but may be challenging to implement), it is unclear how the more complicated areas related to harmonization will be achieved for points 1 (selection of essential targets) and 2 (reagent testing). A detailed process for harmonization and standardization needs to be delineated for recommending targets, selecting optimal antibodies, and leveraging aggregated data.

Regarding recommended targets: In the tables, some antibodies are listed as 'essential' however the authors appropriately highlight the amount of rich information that can be gleaned even from experiments which use only 4-6 antibodies; very valuable experimental studies of these organ tissues can be conducted that do not use many of the listed 'essential' antibodies (thus, one might argue that the term 'essential' is highly dependent on the experimental question/context). Incorrectly used by manuscript reviewers, the existence of essential OMAP panels can become unnecessarily restrictive and actually make experiments more costly for investigators (rather than being a driver of cost savings as proposed). In addition, there will be no doubt significant disagreement about what markers are truly essential for appropriate study of a particular organ. These differences will become even more pronounced as panels are proposed that are not solely focused on providing tissue specific lineage-type and cell type information but also focused on biological processes that assess cell states such as cell signaling, cell death, cell proliferation, immune cell activation and dysfunction, etc. and for subtyping cells that have multiple complex states (e.g., macrophages and the many variants of microglial). The concept of OMAPs as providing recommended lists of useful targets is likely a welcome advance whereas the concept of OMAPs as 'required' targets for a study will likely meet resistance (it is difficult to propose essentiality in the absence of experimental context).

Regarding optimal antibodies: there are practical challenges to harmonization and standardization that need to be further explored with regards to antibodies that will be recommended. For many targets, 2 to 4 different antibody clones are listed in the seven inaugural OMAPs generated by different groups – examples include the antibodies recommended for α SMA, BCL2, CD3, CD8, CD25, CD31, CD34, CD68, CD163, e-cadherin, FOXP3, Ki67, vimentin. It may very well be the case that different clones are needed for different technologies (and for different tissue preparations – FFPE, PFA, or fresh frozen) but there is no sense that cross-platform comparisons were made to assess if indeed there are common 'best' clones or if the presented clones happen to be good enough but not necessarily the very best currently available. Understandably this is the beginning of a long-term effort, but a clear vision of how harmonization and potentially convergence will be achieved, comparisons made (perhaps on shared tissue resources that are commonly imaged?), and decisions made about what constitutes an optimal reagent is important to describe.

Regarding aggregated data: the main text mentions the goal of aggregating usage data across laboratories and technologies (from the broader research community), but it is not clear how that aggregated data will be assessed and what aggregated data the OMAP effort is seeking from the spatial

biology community (primary data seems important to assess the quality of the recommendations – see point 5 below). If more studies/groups report OMAPs using one clone over another, is that the criteria that will be used for the standardized common OMAPs (majority rules)?

Harmonization of aggregated data will become increasingly complex and perhaps the OMAPs will be less unifying than originally envisioned. The current implantation of the OMAPs has captured the antibody preferences of several prominent groups for the particular technologies that they use, however, these OMAPs do not include information about several other widely used methods (mIHC, MIBI, 4i, IMC) so presumably the complexity of the information and divergence will only increase as information from extant and emerging platforms is included. It is easy to envision that there might be “OMAP_Intestine_CODEX,” “OMAP_Intestine_IMC,” “OMAP_Intestine_MIBI,” “OMAP_Intestine_mIHC,” etc. for each of the technologies or perhaps “OMAP_Intestine_CODEX-FFPE,” “OMAP_Intestine_CODEX-Frozen”, etc for different tissue preparation. In addition, will the OMAPs change when platforms change: for example “OMAP_Intestine_CODEX,” and then “OMAP_Intestine_PHENOCYCLER-FUSION”. Will the OMAPs be versioned? These are all important considerations. It even begs the question of "What is an OMAP when there are so many different possibilities?"

4. With the above listed challenges to OMAPs, it is critical to describe the governance structure of the OMAP effort which explains who will evaluate suggested updates, who adjudicates disagreements, who will ensure that this effort lives on past this particular consortium effort, etc.

5. While data tables that collect information on recommended targets and recommended clones is important to share, it is likely going to be very important that key exemplar primary data for each technology/OMAP and processed data are shared with OMAP community – the availability of such data is key so that groups can make independent assessments of the value of the information proposed in the OMAPs.

It feels that the format of a correspondence manuscript is too restrictive to fully address the concerns/questions that are raised here about the proposal for this foundational effort for the emerging spatial biology field.

Minor points:

The main text mentions that on average two clones were evaluated for each target; presumably one antibody was tested for many targets. It would help to know how many targets were evaluated using only one antibody and how many using more than one antibody (ideally a list of all of the antibodies tested for each target would be made available through this resource) – if a user is currently using an antibody that is not recommended in the OMAPs, this list would indicate whether an assessment of that particular reagent had been made and that the antibody had been deemed to have an inferior performance.

The main text lists 178 cell types that can be discerned using the OMAPs; it would help to have those cell types listed in a supplementary table in this manuscript and the marker combinations that are used to define each of those cell types. Cell state lineage dendrograms would be helpful for communicating this information but it can be achieved in table form too. Also, a supplementary table of the 95 anatomical structures that can be identified would also be useful.

In the online tables, the application for each of the methods is listed as IHC which I think should be removed. Some would argue that all of the IF methods used here should not be classified as immunohistochemistry – IHC is a method where an antibody conjugated to an enzyme (e.g., peroxidase or alkaline phosphatase) is used to catalyze a reaction that produces a colored precipitate on the tissue. It is clear that SIMS, however, is certainly not IHC.

As mentioned above, the fields in the online data tables should be cleaned and harmonized (additional examples: sometimes a clone is listed as ‘KP1’ and other times as ‘monoclonal KP1’. In the ‘skin’; OMAP table, the protein targets are labeled starting with “anti-“, when that designation should be made for the antibody but not the target). Structured data is needed.

In the rationale column for CD117 for the Intestine OMAP – “interstitial cells of cajan” should be “interstitial cells of Cajal.” The tables need proofing.

Author Rebuttal to Initial comments

Reviewers' Comments:

Reviewer #1:

Remarks to the Author:

This work is not a traditional research report, but rather an outline of an idea that is underway to produce Organ Mapping Antibody Panels (OMAPs). The concept addresses a key problem in the spatial biology field; the challenge of antibody-based cell identification. OMAP proposes generation and sharing of panels of antibodies to facilitate tissue mapping that can be universally used to define cell types and anatomic structures. This work is the announcement of the first data installment including 204 antibodies that identify 95 structures and 178 cell types.

Overall, this is a great concept. Atlases are useful but can show conflicting data. This proposed approach has the potential to minimize conflict by sharing data, images, methods, and reagents. While time is needed to prove its true value, this introduction seems promising and provocative.

Thank you very much for your enthusiasm for our work. We share your optimism for the value of this offering and appreciate your thoughtful review.

Reviewer #2:

Remarks to the Author: In the correspondence manuscript entitled “Organ Mapping Antibody Panels (OMAPs): A community resource for standardized multiplexed tissue imaging” (NETH-C48483A) an effort is presented for sharing information and standardizing antibody panels for use in multiplexed tissue imaging experiments. The correspondence identifies key challenges in the design and execution of these imaging studies. It appears that there are two key parts to the generation of each particular OMAP: 1. expert selection of targets that are deemed necessary for a deep characterization of a specific tissue type and 2. rigorous antibody testing to identify the optimal reagents to use for each particular technology. While a hub for antibody information and guidance on tissue-specific panels is needed to help overcome some of the barriers for imaging experiments, the current instantiation of the OMAP project raises several concerns.

Thank you very much for your insightful comments and suggestions. We have invested considerable time and effort to address your concerns.

1. More clarity is needed about the criteria and quantitative measures that will be used for optimal reagent selection and rigorous reagent standardization. Review of the manuscript and the OMAP SOP available online does not provide a clear sense of the specific criteria that were used for selecting the antibodies that are included in the current OMAPs (versus the ones that were triaged as having inferior performance). Were similar criteria used by all groups or are the resulting first generation OMAPs based upon very different standards implemented by each of the groups? What criteria should others in the spatial biology community use to suggest changes, additions, improvements, etc. Presumably the best clone was evaluated by visual review of images and also by rigorous quantitative methods that permit the assessment of signal to noise and other features of antibody performance; such data is important for others to review/assess and a detailed process for antibody comparison is needed if the goal is rigorous antibody standardization. It is important to record the types of control tissues and samples that were used in these evaluations. Also, antibody clones may seem to perform differently when assessed with different analytical pipelines (thus, the analysis pipeline used for the assessments should be indicated). Moreover, it seems that only the ‘best’ antibodies that were selected for each OMAP are presented – it is equally important to know which antibodies were evaluated and deemed to be of lesser quality/value for the particular technology/application. Without reporting the specific quantitative criteria and process for selecting optimal antibodies, it is unclear how choices will be made as this resource develops and expands.

We agree that it is important to report the criteria for reagent selection. Since it is not possible to evaluate all available clones for suitability in these assays, even with community efforts, a prioritization of clones is performed using best practices. These practices include evaluation of published results or images, validations provided by antibody databases or community projects (such as the Human Protein Atlas), and direct communication with the antibody vendors or platform developers who fortunately are members of our community. As outlined in our revised manuscript, each antibody included in an OMAP will have an accompanying Antibody Validation Report (AVR). AVRs provide an overview of the antibodies validated for each protein marker, any alternative clones tested, and specific details of the characterization process (initial screening and positive/negative controls, conjugation success, etc). Each AVR will also provide a digital object identifier (DOI) for the validation protocol used for antibody characterization. While the specific workflow may vary by assay and investigator, several clearly defined criteria are shared including qualitative assessment of antibody specificity, sensitivity, and reproducibility. Our goal is to vet antibodies that perform well in a multiplex panel, providing users with a valuable starting point. OMAPs are designed to integrate with AVRs through common metadata fields (Figure 2). We updated our AVR documents to address criticisms and concerns raised here (Supplementary Table 3).

In response to the reviewer's comment, we have provided additional details about antibody validation and the generation of AVRs in the main text and methods, our revised OMAP SOP, and newly created AVR SOP (to be published on Zenodo by December 15th). We agree that it is equally important to capture antibodies that work well but are not included in an OMAP to aid other investigators. Such antibodies are captured by the AVRs under Section IV Validation Data: Other antibodies tested (Supplementary Table 3). Antibodies that are *not* recommended by HuBMAP and *not* included in an OMAP are *not* presently reported in a meaningful way. However, we have many active discussions about creating such a resource for the community, since negative results are equally informative and unfortunately often omitted from scientific publications.

Version control exists for OMAPs via GitHub and Common Coordinate Framework (CCF) releases made every 6 months. In the latest release, we have updated our SOP to request OMAP authors to state the analytical platform(s) used, if any, in the OMAP Description Document that accompanies each OMAP (Supplementary Table 5). These details and the specific quantitative criteria used to select optimal reagents are frequently captured in the publications and protocols accompanying each OMAP.

2. It seems that structured data will be necessary to maintain uniform reporting and information communication in the OMAP data tables. Even with the first seven OMAPs that have been generated by groups that work very closely together in the same consortium, there is a sense

that complexity will quickly arise. Even among a relatively small number of groups, the presentation of the data needs improvement: Smooth muscle actin is called SMA, α -SMA, β -Smooth Muscle Actin. Antibody use is recommended as either a dilution or a specific concentration (ug/ml). In the Skin OMAP, the recommended use is “Dilution/Concentration” and then a number is provided that is presumably the concentration. In the ‘conjugate’ column for the Intestine OMAP, ‘oligonucleotide’ is listed for all antibodies whereas for other OMAPs that also use CODEX a specific fluor is provided (which is the proper way of presenting the information). The rationale for the antibody used is not uniformly presented either (for example CD4 marks both T helper cells as well as macrophages at lower levels presumably in all tissues but that is the listed rationale for only some of the CD4 antibodies but not others). There are even a couple of antibodies that come from an individual investigator’s lab which will not presumably be widely available to others. At the inception it will be important to have more structured ways of reporting and ingesting data otherwise these tables will quickly become difficult to use. Is structured information ingress planned?

We agree with the reviewer and these are actually “teething problems” often encountered in large scale community efforts. It has been a tremendous effort to validate OMAPs and to coordinate such a large number of groups for the first time under this scope. However, as the reviewer pointed out, these harmonization issues, even if minor, reduce the utility of the resource and ease of use. Following the reviewer’s apt remarks, a new version of the tables has been prepared that addresses many of these issues, including a harmonization of protein names and information structure. These details are included in our revised SOP as well as Supplementary Table 4. The appearance of strange characters is due to an individual’s Excel settings. We will provide OMAP Tables as .csv and .xlsx for easy viewing on the portal ([HuBMAP: CCF Portal \(hubmapconsortium.github.io\)](https://hubmapconsortium.github.io)). We also revised our SOP to ask authors to remove improperly rendered characters from antibody names (trademarks, Greek letters). We have expanded our rationales in the OMAP tables and ask all authors to detail the anatomical structures and cell types profiled by each OMAP using guidelines in our SOP (also Supplementary Table 2). With regard to CD4 expression on T helper cells and macrophages, we agree that this is a possibility. We asked OMAP authors to phenotype cells based on the literature and direct observations from images created using the contributed OMAP. Additional harmonization and review of the reported information has been prioritized for all upcoming versions of the resource. Lastly, we are working to automate ingest with the Mapping Component-Indiana University (MC-IU) and HuBMAP Integration, Visualization & Engagement (HIVE) of HuBMAP as suggested. This is a long-term goal.

3. While standardizing data ingress of structured data is ultimately sufficiently straightforward as a concept (but may be challenging to implement), it is unclear how the more complicated areas related to harmonization will be achieved for points 1 (selection of essential targets) and 2

(reagent testing). A detailed process for harmonization and standardization needs to be delineated for recommending targets, selecting optimal antibodies, and leveraging aggregated data.

To achieve this aim, the Affinity Reagent Working Group has partnered with the Anatomical Structures, Cell Types and Biomarkers (ASCT+B) Working Group of HuBMAP. The ASCT+B Working Group is cataloging the diverse anatomical structures, cell types, and gene and protein biomarkers present in human tissues using controlled vocabularies. These data can be explored with a state-of-the-art visualization tool: the ASCT+B Reporter (<https://hubmapconsortium.github.io/ccf-asct-reporter>).

Via the ASCT+B tables, relevant cell types (CTs) and protein biomarkers (BProtein) in these Anatomical Structures (AS) can be retrieved. We are working to expand the ASCT+B table format to link to OMAP data (Figure 2). This will facilitate connecting OMAPs to the ASCT+B tables through their common data type of protein biomarker (BProtein). Once this is completed, output formats will be added to the ASCT+B Reporter to 1) provide the AS, CT and BProtein coverage of a single OMAP, 2) allow comparison of AS, CT, and B shared between OMAPs (Examples: organ, multiplexed antibody imaging methods, and tissue preservation methods), and 3) export visualizations. We are working to extend the ASCT+B Reporter such that protein biomarkers now link to validated antibodies. This way, users can explore what antibodies are validated for which organs, tissues, cell types, proteins, and technologies and export visualizations. In addition, we will add:

- 1) Downloadable output formats that will facilitate authoring new OMAPs by using existing ASCT+B organ table information to provide a list of ranked protein biomarkers based on the number of associated cell types with a particular protein biomarker. This will be helpful for prioritizing certain markers over others in terms of organ- or tissue-specific subsets.
- 2) Drop down menu of Resources linking sample preservation format (FFPE versus fixed frozen) for helping users to identify high usage clones and/or antibodies.
- 3) Feature selection that filters information to include only the terminal anatomical structure (AS) in the ASCT+B table where the Cell type (CT) is located and protein biomarkers (BProtein) characterizing the cell type.

The integration of ASCT+B tables and OMAPs recently received funding from the Common Fund's Cellular Senescence Network (SenNet) and will capitalize on the OMAP initiative described here.

4. Regarding recommended targets: In the tables, some antibodies are listed as 'essential' however the authors appropriately highlight the amount of rich information that can be gleaned

even from experiments which use only 4-6 antibodies; very valuable experimental studies of these organ tissues can be conducted that do not use many of the listed 'essential' antibodies (thus, one might argue that the term 'essential' is highly dependent on the experimental question/context). Incorrectly used by manuscript reviewers, the existence of essential OMAP panels can become unnecessarily restrictive and actually make experiments more costly for investigators (rather than being a driver of cost savings as proposed). In addition, there will be no doubt significant disagreement about what markers are truly essential for appropriate study of a particular organ. These differences will become even more pronounced as panels are proposed that are not solely focused on providing tissue specific lineage-type and cell type information but also focused on biological processes that assess cell states such as cell signaling, cell death, cell proliferation, immune cell activation and dysfunction, etc. and for subtyping cells that have multiple complex states (e.g., macrophages and the many variants of microglial). The concept of OMAPs as providing recommended lists of useful targets is likely a welcome advance whereas the concept of OMAPs as 'required' targets for a study will likely meet resistance (it is difficult to propose essentiality in the absence of experimental context). This is a valid concern. We have removed the "essential" column from the OMAP Table. As outlined in our revised SOP, we ask OMAP authors to state if a particular marker represents a prioritized target. This will enable users not familiar with the organ system to start their potential target review from those that the OMAP teams, in coordination with the ASCT-B group mentioned above, found as most useful for their work. This helps guide user attention without being restrictive, since in diverse investigations other targets might be more useful or just a small subset might be necessary. Our updated SOP includes the following statement that we believe addresses this concern: "Beyond the 4-6 core markers, the OMAP will include antibodies that facilitate biological exploration of the specified organ. At a minimum, an OMAP should allow 10 or more unique biomarkers to be visualized in a single tissue section. An ideal panel will be as comprehensive as possible (20-60+ markers as technologies advance). We ask experts to select antibodies that capture the greatest number of cell types and anatomical structures based on their domain knowledge, survey of the literature, consultation with pathologists, and review of the ASCT+B table for their organ."

5. Regarding optimal antibodies: there are practical challenges to harmonization and standardization that need to be further explored with regards to antibodies that will be recommended. For many targets, 2 to 4 different antibody clones are listed in the seven inaugural OMAPs generated by different groups – examples include the antibodies recommended for aSMA, BCL2, CD3, CD8, CD25, CD31, CD34, CD68, CD163, e-cadherin, FOXP3, Ki67, vimentin. It may very well be the case that different clones are needed for different technologies (and for different tissue preparations – FFPE, PFA, or fresh frozen) but there is no sense that cross-platform comparisons were made to assess if indeed there are common 'best' clones or if the presented clones happen to be good enough but not necessarily

the very best currently available. Understandably this is the beginning of a long-term effort, but a clear vision of how harmonization and potentially convergence will be achieved, comparisons made (perhaps on shared tissue resources that are commonly imaged?), and decisions made about what constitutes an optimal reagent is important to describe.

Thank you again for your detailed review of our 7 data tables. As outlined in point #1, we are integrating AVR_s with OMAP_s to facilitate the identification of optimal reagents across HuBMAP, SenNet, and other participating consortia. Integration of the ASCT+B Reporter with OMAP_s is our solution for achieving harmonization and potential convergence of recommended clones across different technologies and tissue preparations (please see point #3). To address the reviewer's comment, we have provided a table of clones used across the 7 OMAP_s (Supplementary Table 1). We do wish to emphasize that OMAP_s are "modular" and that other members of the consortium are expanding upon several of these first 7 OMAP_s for their tissue and platform of interest. This practice will certainly result in greater overlap between clones and antibodies across users. In parallel, we are working with industry leaders to create commercial OMAP_s that will greatly facilitate standardization and harmonization. We have discussed how to compare antibody performance across HuBMAP Tissue Mapping Centers (TMCs) using a shared tissue source as suggested. However, these conversations have not materialized into concrete action at this time. Lastly, this is a community effort and as more researchers join, more reagents will be validated, harmonized, and extended to other imaging modalities and tissues. This is the unmatched value of community efforts, since they can go well beyond what a single group or grant can achieve.

6. Regarding aggregated data: the main text mentions the goal of aggregating usage data across laboratories and technologies (from the broader research community), but it is not clear how that aggregated data will be assessed and what aggregated data the OMAP effort is seeking from the spatial biology community (primary data seems important to assess the quality of the recommendations – see point 5 below). If more studies/groups report OMAP_s using one clone over another, is that the criteria that will be used for the standardized common OMAP_s (majority rules)?

We apologize for the potential miscommunication. As mentioned in point #5, we are partnering with antibody manufacturers and industry leaders in spatial biology to construct commercial OMAP_s. All vendors have internal validation and conjugation records and have offered to support us by providing these resources. Furthermore, we will identify publications from highly multiplexed studies with accompanying primary data. We plan to solicit OMAP authorship from these individuals in 2023. In general, we agree with a potential majority rule. In fact, many OMAP authors already engage in this practice by identifying highly cited antibodies through vendors and antibody search engines. However, this of course doesn't mean that the majority is

always right. This is why we have decided to adhere to strict internal validation criteria that are uniformly applied, even for well-established clones.

7. Harmonization of aggregated data will become increasingly complex and perhaps the OMAPs will be less unifying than originally envisioned. The current implantation of the OMAPs has captured the antibody preferences of several prominent groups for the particular technologies that they use, however, these OMAPs do not include information about several other widely used methods (mIHC, MIBI, 4i, IMC) so presumably the complexity of the information and divergence will only increase as information from extant and emerging platforms is included. It is easy to envision that there might be “OMAP_Intestine_CODEX,” “OMAP_Intestine_IMC,” “OMAP_Intestine_MIBI,” “OMAP_Intestine_mIHC,” etc. for each of the technologies or perhaps “OMAP_Intestine_CODEX-FFPE,” “OMAP_Intestine_CODEX-Frozen”, etc for different tissue preparation. In addition, will the OMAPs change when platforms change: for example “OMAP_Intestine_CODEX,” and then “OMAP_Intestine_PHENOCYCLER-FUSION”. Will the OMAPs be versioned? These are all important considerations. It even begs the question of "What is an OMAP when there are so many different possibilities?"

We have actively solicited OMAPs from experts employing 4i, MIBI, IMC, and have vendor support from Miltenyi for their MACSima platform. We greatly appreciate your insights into naming and have updated our SOP and manuscript to include a new naming convention that takes into consideration the date/version, tissue, imaging platform, and sample preservation method, e.g., OMAP-1 (Figure 2). We have updated the names of the 7 OMAPs on the portal to address these issues (new release December 15th, 2022). Answers to such questions as “what is an OMAP?” and “will OMAPs be versioned?” are included in our FAQs.

8. With the above listed challenges to OMAPs, it is critical to describe the governance structure of the OMAP effort which explains who will evaluate suggested updates, who adjudicates disagreements, who will ensure that this effort lives on past this particular consortium effort, etc. Thank you very much. We address this issue in the Governance section of the Methods.

9. While data tables that collect information on recommended targets and recommended clones is important to share, it is likely going to be very important that key exemplar primary data for each technology/OMAP and processed data are shared with OMAP community – the availability of such data is key so that groups can make independent assessments of the value of the information proposed in the OMAPs.

We absolutely agree with this point and outline the following plan to provide meaningful visualizations and examples of OMAP data:

1. Short-term: on the OMAP landing page, add a link to an existing dataset/image for each OMAP

- a. OMAPs authored by HuBMAP members: link to a representative image within the HuBMAP Data Portal
 - b. Other OMAPs: provide an alternate DOI (e.g., for manuscript or image in a persistent data repository such as Zenodo)
 - c. Please see revised SOP for details and OMAP Description Document template (Supplementary Table 5).
2. Longer-term: deploy the Vitessce image viewer separately from the Data Portal to specifically highlight OMAPs
 - a. As part of OMAP submission, authors will provide a representative image and corresponding annotations that specify key anatomical structures and/or functional tissue units.
 - b. Vitessce development team will assist with workflow to (a) display annotations as an overlay on the imaging data and (b) enable grouping/labeling of channels in such a way that visually links biomarkers to cell types and/or anatomical structures.
 - c. Please see Figure 2 for future data integration plans using Vitessce.

Our intention is to ultimately tie key components of the OMAP into existing data structures: each antibody clone will be associated with an AVR, while each biomarker and cell type(s) or anatomical structure(s) would be associated with the ASCT+B table for that particular organ. In this way, OMAP consumers will have all data at their fingertips to make informed decisions between OMAPs or determine their own desired additions/modifications. Our envisioned workflow and future data integration plans are outlined in Figure 2.

10. It feels that the format of a correspondence manuscript is too restrictive to fully address the concerns/questions that are raised here about the proposal for this foundational effort for the emerging spatial biology field.

We agree and are grateful the editor allowed us to expand our Correspondence into a Brief Communication.

11. Minor points:

The main text mentions that on average two clones were evaluated for each target; presumably one antibody was tested for many targets. It would help to know how many targets were evaluated using only one antibody and how many using more than one antibody (ideally a list of all of the antibodies tested for each target would be made available through this resource) – if a user is currently using an antibody that is not recommended in the OMAPs, this list would indicate whether an assessment of that particular reagent had been made and that the antibody had been deemed to have an inferior performance.

The reviewer raises very important points about the intricacies of antibody validation and selection. We have been discussing how best to capture information about antibodies that were tested but not used, in part because there are many reasons a clone might not have been included in the OMAP. This is not limited to antibody failure in a specific application; cost, commercial conjugate availability, isotype, or antigen effects are also contributing factors (Figure 2). We are working to (a) streamline AVR data structure while still keeping it accessible and incentivizing its adoption and (b) develop a framework to link biomarker targets with AVRs and ASCT+B tables (see responses above). We are also actively discussing about capturing and providing as much information as possible about failed clones and the exact issue(s) identified (e.g. failed CODEX conjugation, low specificity in FFPE, etc), which as we mentioned can often save researchers as much time and effort as positive results. Our revised AVR template will include this information under Section IV Validation Data: Other antibodies tested (Supplementary Table 3).

12. The main text lists 178 cell types that can be discerned using the OMAPs; it would help to have those cell types listed in a supplementary table in this manuscript and the marker combinations that are used to define each of those cell types. Cell state lineage dendrograms would be helpful for communicating this information but it can be achieved in table form too. Also, a supplementary table of the 95 anatomical structures that can be identified would also be useful.

As requested, we added a supplementary table detailing the anatomical structures, cell types, and marker combinations for each OMAP (Supplementary Table 2). Anatomical structures and cell types are based on the ASCT+B tables for each organ and extended using literature and domain knowledge. Whenever possible, we used standardized Cell Ontology (CL) terms to denote cell phenotypes. We also highlighted core markers for each cell type as a resource for identifying high priority targets. As highlighted in our revised text and Figure 2, full integration with the ASCT+B reporter is a long-term goal but not feasible here.

13. In the online tables, the application for each of the methods is listed as IHC which I think should be removed. Some would argue that all of the IF methods used here should not be classified as immunohistochemistry – IHC is a method where an antibody conjugated to an enzyme (e.g., peroxidase or alkaline phosphatase) is used to catalyze a reaction that produces a colored precipitate on the tissue. It is clear that SIMS, however, is certainly not IHC. We have removed all instances of “IHC” from our revised OMAP Tables.

14. As mentioned above, the fields in the online data tables should be cleaned and harmonized (additional examples: sometimes a clone is listed as ‘KP1’ and other times as ‘monoclonal KP1’. In the ‘skin’; OMAP table, the protein targets are labeled starting with “anti-“, when that designation should be made for the antibody but not the target). Structured data is needed.

We agree and are, again, grateful for your thorough review. We have updated our SOP to aid other users in filling out the metadata in a standardized way to avoid such errors in the future. All tables have been corrected and updated on the portal (live December 15th, 2022).

15. In the rationale column for CD117 for the Intestine OMAP – “interstitial cells of cajan” should be “interstitial cells of Cajal.” The tables need proofing.

Thank you. All OMAP tables have been reviewed and proofed by original OMAP authors and 6 external reviewers.

Decision Letter, first revision:

4th Jan 2023

Dear Andrea,

Happy New Year!

Thank you for submitting your revised manuscript "Organ Mapping Antibody Panels (OMAPs): A community resource for standardized multiplexed tissue imaging" (NMETH-BC48483B). It has now been seen by the original referees and their comments are below. The reviewers find that the paper has improved in revision, and therefore we'll be happy in principle to publish it in Nature Methods, pending minor revisions to satisfy the referees' final requests and to comply with our editorial and formatting guidelines.

TRANSPARENT PEER REVIEW

Nature Methods offers a transparent peer review option for new original research manuscripts submitted from 17th February 2021. We encourage increased transparency in peer review by publishing the reviewer comments, author rebuttal letters and editorial decision letters if the authors agree. Such peer review material is made available as a supplementary peer review file. Please state in the cover letter 'I wish to participate in transparent peer review' if you want to opt in, or 'I do not wish to participate in transparent peer review' if you don't. Failure to state your preference will result in delays in accepting your manuscript for publication.

Please note: we allow redactions to authors' rebuttal and reviewer comments in the interest of confidentiality. If you are concerned about the release of confidential data, please let us know

specifically what information you would like to have removed. Please note that we cannot incorporate redactions for any other reasons. Reviewer names will be published in the peer review files if the reviewer signed the comments to authors, or if reviewers explicitly agree to release their name. For more information, please refer to our [FAQ page](https://www.nature.com/documents/nr-transparent-peer-review.pdf).

ORCID

Sincerely,
Madhura

Madhura Mukhopadhyay, PhD
Senior Editor
Nature Methods

Reviewer #2 (Remarks to the Author):

The authors have provided a thoughtful response. The expanded and edited text, additional figure, supplementary tables, and revised/new SOPs address my prior concerns and the response discusses both short and long term future goals for this community resource and how to achieve them.

Final Decision Letter:

Dear Andrea,

I am pleased to inform you that your Brief Communication, "Organ Mapping Antibody Panels (OMAPs): A community resource for standardized multiplexed tissue imaging", has now been accepted for

publication in Nature Methods. Your paper is tentatively scheduled for publication in our May print issue, and will be published online prior to that. The received and accepted dates will be 12 Jul, 2022 and 14 Mar, 2023. This note is intended to let you know what to expect from us over the next month or so, and to let you know where to address any further questions.

Over the next few weeks, your paper will be copyedited to ensure that it conforms to Nature Methods style. Once your paper is typeset, you will receive an email with a link to choose the appropriate publishing options for your paper and our Author Services team will be in touch regarding any additional information that may be required.

Your paper will now be copyedited to ensure that it conforms to Nature Methods style. Once proofs are generated, they will be sent to you electronically and you will be asked to send a corrected version within 24 hours. It is extremely important that you let us know now whether you will be difficult to contact over the next month. If this is the case, we ask that you send us the contact information (email, phone and fax) of someone who will be able to check the proofs and deal with any last-minute problems.

If, when you receive your proof, you cannot meet the deadline, please inform us at rjsproduction@springernature.com immediately.

Once your manuscript is typeset and you have completed the appropriate grant of rights, you will receive a link to your electronic proof via email with a request to make any corrections within 48 hours. If, when you receive your proof, you cannot meet this deadline, please inform us at rjsproduction@springernature.com immediately.

Once your paper has been scheduled for online publication, the Nature press office will be in touch to confirm the details.

Content is published online weekly on Mondays and Thursdays, and the embargo is set at 16:00 London time (GMT)/11:00 am US Eastern time (EST) on the day of publication. If you need to know the exact

publication date or when the news embargo will be lifted, please contact our press office after you have submitted your proof corrections. Now is the time to inform your Public Relations or Press Office about your paper, as they might be interested in promoting its publication. This will allow them time to prepare an accurate and satisfactory press release. Include your manuscript tracking number NMETH-BC48483C and the name of the journal, which they will need when they contact our office.

About one week before your paper is published online, we shall be distributing a press release to news organizations worldwide, which may include details of your work. We are happy for your institution or funding agency to prepare its own press release, but it must mention the embargo date and Nature Methods. Our Press Office will contact you closer to the time of publication, but if you or your Press Office have any inquiries in the meantime, please contact press@nature.com.

If you are active on Twitter, please e-mail me your and your coauthors' Twitter handles so that we may tag you when the paper is published.

Please note that *Nature Methods* is a Transformative Journal (TJ). Authors may publish their research with us through the traditional subscription access route or make their paper immediately open access through payment of an article-processing charge (APC). Authors will not be required to make a final decision about access to their article until it has been accepted. [Find out more about Transformative Journals](https://www.springernature.com/gp/open-research/transformative-journals)

To assist our authors in disseminating their research to the broader community, our SharedIt initiative provides you with a unique shareable link that will allow anyone (with or without a subscription) to read

the published article. Recipients of the link with a subscription will also be able to download and print the PDF. As soon as your article is published, you will receive an automated email with your shareable link.

Please note that you and your coauthors may order reprints and single copies of the issue containing your article through Springer Nature Limited's reprint website, which is located at <http://www.nature.com/reprints/author-reprints.html>. If there are any questions about reprints please send an email to author-reprints@nature.com and someone will assist you.

Best regards,
Madhura

Madhura Mukhopadhyay, PhD
Senior Editor
Nature Methods